# Detection of Forest Fires through Deep Unsupervised Learning Modeling of Sentinel-1 Time Series

Thomas Di Martino [1,2,*], Bertrand Le Saux [3], Régis Guinvarc'h [1], Laetitia Thirion-Lefevre [1] and Elise Colin [2]

1   SONDRA, CentraleSupélec, Université Paris-Saclay, 3 Rue Joliot Curie, 91190 Gif-sur-Yvette, France; regis.guinvarch@centralesupelec.fr (R.G.); laetitia.thirion@centralesupelec.fr (L.T.-L.)
2   DTIS, ONERA, Université Paris-Saclay, 91123 Palaiseau, France; elise.colin@onera.fr
3   Φ-Lab, European Space Agency (ESA), 00044 Frascati, Italy; bertrand.le.saux@esa.int
*   Correspondence: thomas.di-martino@centralesupelec.fr

**Abstract:** With an increase in the amount of natural disasters, the combined use of cloud-penetrating Synthetic Aperture Radar and deep learning becomes unavoidable for their monitoring. This article proposes a methodology for forest fire detection using unsupervised location-expert autoencoders and Sentinel-1 SAR time series. The models are trained on SAR multitemporal images over a specific area using a reference period and extract any deviating time series over that same area for the test period. We present three variations of the autoencoder, incorporating either temporal features or spatiotemporal features, and we compare it against a state-of-the-art supervised autoencoder. Despite their limitations, we show that unsupervised approaches are on par with supervised techniques, performance-wise. A specific architecture, the fully temporal autoencoder, stands out as the best-performing unsupervised approach by leveraging temporal information of Sentinel-1 time series using one-dimensional convolutional layers. The approach is generic and can be applied to many applications, though we focus here on forest fire detection in Canadian boreal forests as a successful use case.

**Keywords:** Synthetic Aperture Radar; forest monitoring; unsupervised learning; remote sensing time series; fire detection

## 1. Introduction

Remote sensing data are critical assets for protecting and improving life on Earth. Historically, a significant milestone regarding Earth Observation for environmental monitoring was set with the launch of the two European Remote Sensing Satellites (ERS-1 and ERS-2) [1], which induced the development of various environmental monitoring methodologies [2–4]. The launch of ERS satellites was then followed by the Environmental Satellite (EnviSat) [5], which itself led to a better understanding of the Earth's mechanisms [6–8]. Recently, a new suite of ESA-branded satellites, the Sentinels [9], is taking over the older missions. They are developed to fill the gaps with former discontinued missions and bring new and more modern technology, focusing on land, ocean, and atmospheric monitoring. A common denominator to the ERS, EnviSat, and Sentinel missions is including a Synthetic Aperture Radar sensor [10]. These active radar sensors allow for continuous imaging, no matter the time of the day or the weather condition. Thanks to their regularity in data quality, plenty of studies leveraged SAR for vegetation monitoring, since ERS-1 SAR sensor [11], through EnviSat ASAR instrument [12], and until the latest Sentinel-1 satellites [13]. The downstream applications of SAR imagery to vegetation are numerous: in agriculture, they are used to classify crop fields [14], monitor crop growth [15], and predict crop yield [16]. For the field of forestry, SAR data has proven essential for retrieving forest biomass [17] and the estimation of forest cover [18].

With a short revisit time (6 to 12 days) and a spatial resolution of $5 \times 20$ m, the launch of Sentinel-1 marked a transition to a new age of applications relying on multitemporal SAR data. Various studies pointed out the seasonality component of the forest in Sentinel-1 time series as a distinctive signature, which can then be used to extract forest types, growing stock volume, or tree cover, among other forest parameters [19–21]. However, constantly monitoring forested sites remains challenging, particularly regarding handling large areas and prolonged periods. In particular, the success of such applications requires state-of-the-art algorithms. For that matter, Deep Learning algorithms have proven themselves as powerful automated tools. Applications such as forest mapping [22,23] and deforestation monitoring [13,24] have greatly benefited from such methods. However, these studies rely on the use of labels for the training of supervised learning algorithms. In the case of novelty detection, critical for monitoring areas of interest, unsupervised models should be considered, as they are trained solely on data without the need for any labels. Unsupervised learning for modeling SAR time series is a new field, with many stones left unturned: it has been proven to be helpful for flood detection [25,26], agricultural monitoring [27–29], and fire detection [30]. Its advantages over supervised learning are numerous: first, obtaining relevant and quality labels for training purposes is a difficult task, as they need to be of a certain degree of quality, but also quantity, to ensure generalization properties of the learned models. Additionally, the supervised learning models suffer from issues related to domain shift [31], degrading their performance over time. Unsupervised learning approaches encapsulate more robust generalization properties, making them more resistant to domain shifts. A commonly used unsupervised architecture is the autoencoder, which models the data distribution. Variations from the modeled distribution are considered either novelties or anomalies, depending on the context. Thus, in this paper, we aim to merge the findings in both the temporal modeling of Forest through C-Band SAR multitemporal data with the latest advances in unsupervised learning for SAR time series to design a ready-to-use methodology for monitoring an area of interest, subject to forest fires. In particular, we present the use of autoencoders to model the yearly Sentinel-1 temporal signature of a forest of interest. We then illustrate how the successful modeling of this signature allows for detecting forest fires by extracting deviations from the modeled expectation. For that matter, we introduce three separate methodologies to the problem of SAR time series autoencoding and compare them together, as well as against standard supervised approaches.

## 2. Methods

We first describe our autoencoder-based pipeline for anomaly detection in monitored areas, then introduce several variants of autoencoders adapted to geospatial time series: Fully Temporal Autoencoder (FTAE), Coupled Spatiotemporal Autoencoder (CSTAE), and Decoupled Spatiotemporal Autoencoder (DCSTAE).

### 2.1. General Methodology

Synthetic Aperture Radar data is vital for modeling the seasonal variations of boreal forests [32]. In particular, the Sentinel-1 satellites have many advantages for monitoring boreal forests, making usable acquisitions regardless of weather conditions. Combined with their high acquisition frequency (6 to 12 days) and open science availability, it makes them an ideal candidate for Boreal Forests monitoring, which are ecosystems known to suffer from cloud cover for a significant portion of the year. In this article, we propose a method for fire monitoring a specific forested Area of Interest (AOI) through the union of SAR multitemporal images and various autoencoder architectures.

Autoencoders are a specific class of deep neural networks trained without supervision. Initially introduced by [33] as an equivalent to a non-linear Principal Component Analysis, its primary objective is twofold: first, to project the input data onto a lower-dimensional plane, and second, to reconstruct the original input using the information from its projected

representation. Given an input $X \in \mathbb{R}^n$, the forward pass of an autoencoder can be written as follows:

$$AE(X) = dec(enc(X; \theta_{enc}); \theta_{dec}) = \tilde{X}$$

where:

- $enc : \mathbb{R}^n \to \mathbb{R}^t$ serves as the encoder, responsible for transforming the input data X into a lower-dimensional latent representation ($n > t$). It usually consists of a stack of operation layers, convolutional or fully connected, interlaced with non-linear activation functions.
- $dec : \mathbb{R}^t \to \mathbb{R}^n$ serves as the decoder, charged with reconstructing the original input using $X$'s latent representation created by $e$. In the same fashion as the encoder, it is also non-linear through operation and activation layers combined.
- $\tilde{X}$ is the reconstructed version of input $X$.
- $\theta_{enc}$, and $\theta_{dec}$ are the respective weight parameters of the encoder and decoder.

The autoencoder fits its weights $\theta_{enc}$, and $\theta_{dec}$ by employing backpropagation to minimize the mean square error, as defined in: $MSE(X, \tilde{X}) = \frac{1}{n} \sum_{i=1}^{n} (X^{(i)} - \widetilde{X^{(i)}})^2$. This error is calculated between the input vector and its corresponding reconstructed version. Through this process, the unsupervised network learns to minimize information loss and eliminate potential noise from the reconstruction. Iteratively, the network learns to model and fit the training data distribution. Thus, it makes autoencoders powerful tools for detecting out-of-distribution samples. In particular, we leverage these capacities as shown in Figure 1.

**Figure 1.** Abstract representation of the proposed methodology: training of an AOI-expert autoencoder on SAR multitemporal images of a reference period, and application of the trained model to extract disturbances in SAR multitemporal images of a monitored period.

The main intuition behind the design consists of two steps, as shown in Figure 1:

1. We use a given reference period where the AOI is undisturbed. An autoencoder neural network is trained on the SAR time series of the area acquired during this period. They then encapsulate the expected temporal signature of the undisturbed forested site.

2. We apply the trained model to the SAR time series of the test period to extract anomalies and deviations from the previously introduced expected temporal signature.

While the first step is straightforward, as it relies on training an autoencoder architecture using the previously introduced objective, the anomaly extraction step is rather more subtle. It relies on the reconstruction error, driving the convergence of the model. We design a complete anomaly extraction pipeline presented in Figure 2.

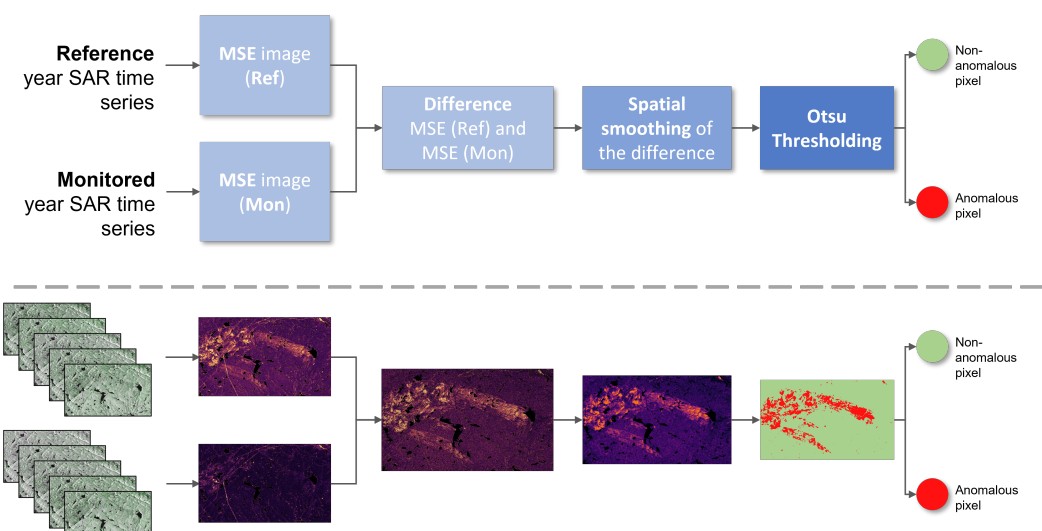

**Figure 2.** Anomaly extraction pipeline.

This pipeline relies on the computation of the MSE of the time series of both the reference and the monitored period. The reference period's MSE, noted $MSE_{ref}$, is considered a baseline of how well the autoencoder can encode and decode the time series of the AOI without disturbances. The monitored period's MSE, noted $MSE_{mon}$, measures the difficulty of the autoencoder to perform the same task due to the presence of fires. The computation of both MSEs is crucial: elements within the AOI can be, by nature, hard to represent by the autoencoder, resulting in generally high MSE in both the reference and monitored time series. If the method only leverages $MSE_{mon}$, these hard-to-model elements would be confused with fires, raising the false alarm rate. Regularizing $MSE_{mon}$ using $MSE_{ref}$ helps to alleviate these hard elements. For that, a difference image is computed, noted $\Delta MSE = MSE_{mon} - MSE_{ref}$. This difference image is then spatially smoothed to increase the retrieval quality, using a 10-by-10 averaging window over the image, to obtain $\Delta MSE_{smoothed}$. Automatic thresholding of the resulting product is then performed to separate anomalous from non-anomalous pixels, using Otsu thresholding [34].

This automatic thresholding strategy aims to separate the spatially smoothed difference image in two groups of pixels: the group (0), corresponding to stable areas, and the group (1) with higher reconstruction error, corresponding to presumed forest fires. For that, the Otsu approach aims to find a threshold $t$ that minimizes the intra-class variance of each group. Practically speaking, this translates in the minimization of this equation: $\sigma_w^2(t) = w_0(t)\sigma_0^2(t) + w_1(t)\sigma_1^2(t)$ where $w_0(t)$ (resp. $w_1(t)$) correspond to the percentage of pixels respecting $\Delta MSE_{smoothed} < t$ (resp. $\Delta MSE_{smoothed} > t$), and $\sigma_0^2(t)$ their variance. The minimization of $\sigma_w^2(t)$ is found using a brute force approach by iterating over the partitioned and sorted set of values of $\Delta MSE_{smoothed}$.

This whole pipeline relies on efficient modeling of the input time series performed by autoencoders. Thus, their design is crucial to the success of the task. We explored and compared three approaches to encoding the AOI time series: a fully temporal approach, a coupled spatiotemporal approach, and a decoupled spatiotemporal approach.

## 2.2. Detailed Autoencoders Architectures

In Figure 3, we present the three benchmarked autoencoder architectures. We first introduce a 1D-CAE, and then, given the problem at hand, we study the potential benefits of adding spatial information to the encoding of SAR time series through two separate approaches.

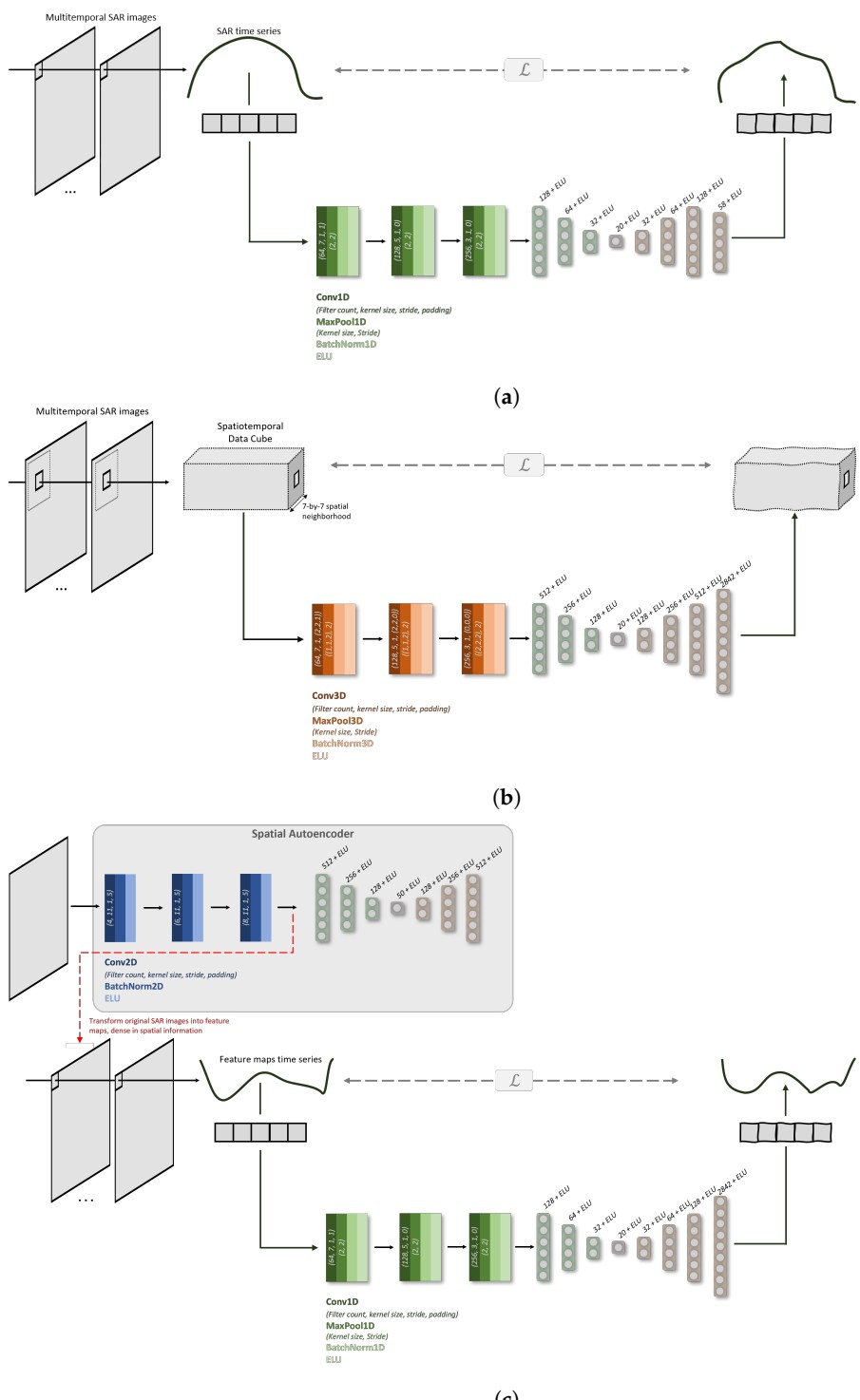

**Figure 3.** The various autoencoding architectures introduced and tested: (**a**) Fully Temporal Autoencoder (FTAE) (**b**) Coupled Spatiotemporal Autoencoder (CSTAE) (**c**) Decoupled Spatiotemporal Autoencoder (DCSTAE).

### 2.2.1. Fully Temporal Autoencoder (FTAE)

Presented in Figure 3a, the temporal autoencoder consists of an encoder and a decoder. The encoder is made of three successive stacks of 1D Convolutions to extract temporal features, 1D Max-pooling [35] to gather the strongest temporal activations, a 1D batch normalization layer [36], to improve convergence, and we finally opt for an Exponential Linear Unit (ELU) [37] activation layer. The temporal features are then fed to three linear layers, each combined with an ELU activation layer, before mapping onto the embedding layer. The original input time series is then progressively reconstructed using the decoder part of the network, which consists of four fully connected layers.

Such a model has already shown remarkable performance in agricultural modeling of SAR time series [27,38]. Its application to forest SAR time series is still to be studied.

### 2.2.2. Coupled Spatiotemporal Autoencoder (CSTAE)

Another approach to the encoding of SAR time series regards considering the backscatter coefficient of neighboring pixels. Additional information regarding the pixel at hand may lie within adjacent pixels. Thus their incorporation into the encoding of the time series sounds legitimate. We developed a coupled spatiotemporal autoencoder to assess this idea, illustrated in Figure 3b. The data preparation for this method differs from the fully temporal approach as we opt for 3D data cubes centered on the pixel of interest, including its neighbors. We designed the model such that it accepts a 7-by-7 neighborhood. This design was empirically found to be pertinent enough to capture neighboring information for a given pixel and not solely consist of speckle noise while not increasing the computational burden significantly. The autoencoder model is then similar to the aforementioned fully temporal autoencoder, with the difference being that all formerly 1D operations become 3D. Thus, we combine three stacks of 3D convolution, 3D max-pooling, and 3D Batch Normalization. A difference regards the max-pooling operation. We use asymmetrical max-pooling for the first two layers to only reduce the temporal dimension through this operation. The spatial dimension is too small to be iteratively reduced in the same way as the temporal dimension. Thus, the spatial resolution is kept through all convolutional layers of the network. After extracting spatiotemporal features, we project them onto the bottlenecking layer of dimension 20 using three fully connected layers combined with ELU activations. The dimensions of the fully connected layers are higher than the fully temporal approach to account for the higher dimensionality of the extracted features.

### 2.2.3. Decoupled Spatiotemporal Autoencoder (DSTAE)

Following the same motivation as the coupled spatiotemporal autoencoder, this approach leverages spatial and temporal information of SAR time series separately, as displayed in Figure 3c. We first train a fully spatial autoencoder on patches of 128-by-128 pixels with no temporal information: acquisitions at different dates are considered as different images and thus result in different patches. We design the spatial autoencoder using combinations of 2D convolutions, 2D batch normalization, and ELU activation functions. Fully connected layers follow in the same fashion as the other two networks. The absence of pooling layers is explained by the usage made of this spatial autoencoder. After training, we apply it to the initial full-sized images, transforming them into feature maps dense in spatial information. We then stack these newly created feature maps temporally and extract pixel-wise time series to train a fully temporal autoencoder with the same architecture as the one mentioned earlier. The time series in question are no longer SAR time series but rather time series of spatial features. Their encoding thus includes both spatial information from the spatial autoencoder and temporal information from the temporal variations of the feature maps.

## 3. Experimental Settings

### 3.1. Study Case

To assess the viability of the developed methodology and how each autoencoder architecture performs, we have focused on the Labrieville forest fire in Québec Province, Canada.

The ground truth fire outline, shown in Figure 4, originates from the National Burned Area Composite (NBAC) of the Canadian Wildland Fire Information System (CWFIS) [39]. The provided fire outline was generated by human sketch mapping using a combination of RapidEye's imagery and airborne images taken from a helicopter. While being spaceborne, the RapidEye constellation offers high-resolution optical for fire monitoring, with a ground-sampling distance of 6.5 m, resampled to 5 m on orthorectified products, making the precision of the fire outline on par with modern open source satellites.

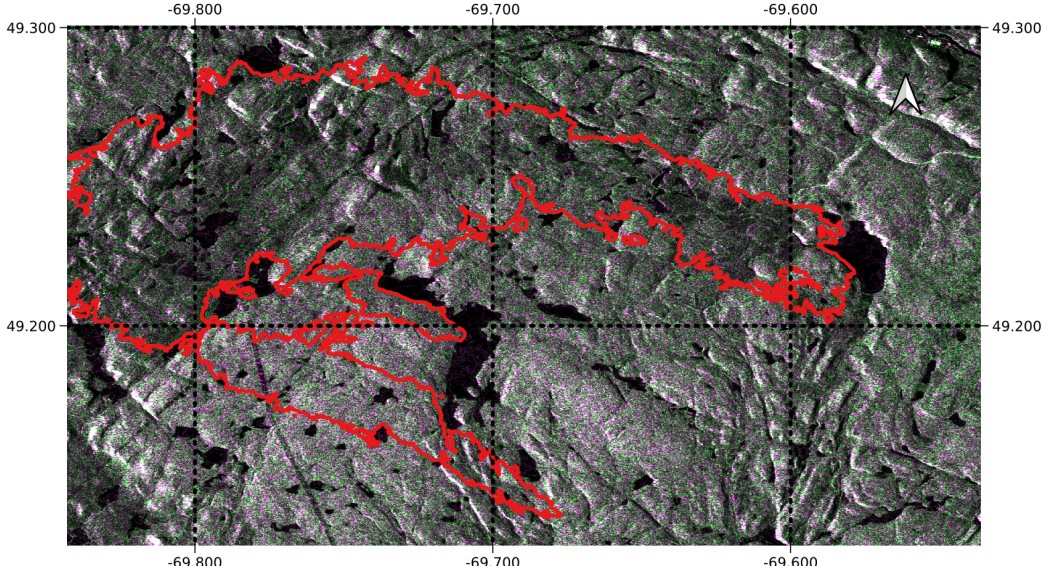

**Figure 4.** Labrieville forest fire outline, with background Sentinel−1 image, acquired on 9 January 2017, displayed in false color (Red: VV, Green: VH, Blue: VV).

Presumably originating from non-industrial human activity, such as campfires, the fire was identified on 26 June 2018 by firefighters who quickly classified it as out of control. The total spread area of the forest fire can be visualized in Figure 5. The fire was ultimately contained on 3 July 2018, and became officially under control three weeks later, although the official fire end date reported in the NBAC data is 28 August 2018. It fully burned a total of 12,986 ha of forest.

To assess the retrieval of this fire using our methodology, we use two years of Sentinel-1 Ground Range Detection (GRD) data for 2017 and 2018 over the fire region. This resulted in 29 acquisitions for 2017, the reference period, and 29 acquisitions for 2018, the monitored period. The Sentinel-1 GRD data comes from Google Earth Engine and consists of the backscattering coefficient in log-scaled intensity for VV and VH polarizations. The acquisitions were made in ascending mode and are part of the 164th orbit. This dataset thus comprises 58 images of 3424-by-2037 pixels, with two polarizations.

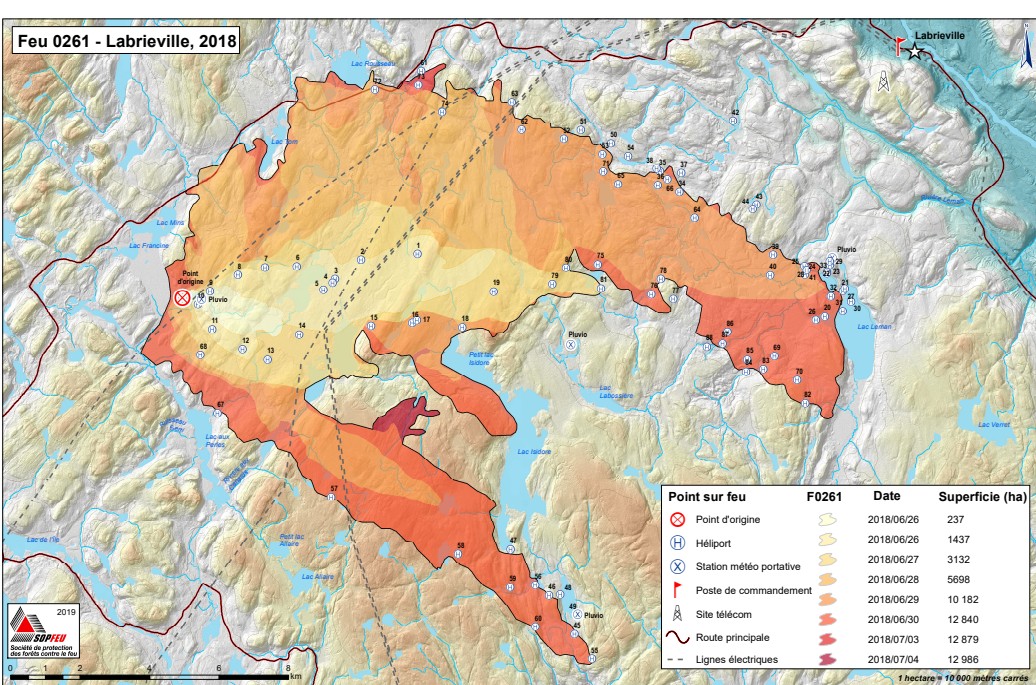

**Figure 5.** Labrieville forest fire map (source: ESRI Map Calendar 2020, SOPFEU).

## 3.2. Evaluation Scheme

The objectives of the experiments are multiple:

- Firstly, to validate the unsupervised approach to the problem, we perform an empirical performance comparison between our methods and classical supervised approaches to position ourselves.
- Secondly, we aim to compare the various proposed philosophies regarding encoding temporal information from SAR multitemporal images and whether adding spatial knowledge is relevant.
- Lastly, we study the impact of the key of the functioning of this method: the dimension of the bottlenecking layer. We aim to explore the impact various dimensions have on the overall performance and tradeoffs that may appear.

As presented, the unsupervised method consists of training on 2017's data and running inference on 2018's. We perform these tests five times for each available methodology, with various embedding sizes. For the supervised methods, however, we train them solely on 2018's data, using a four-fold cross-validation approach, and we repeat the training five times for statistical relevance of the measured performance.

For that matter, we aim to evaluate our methodology using separate metrics, which aim at measuring the behavior of the methodologies through different lenses.

A first set of four metrics, dealing with binary predictions, are used:

- The accuracy, which provides an oversight of the correctness of the predictions but may also be sensible to class imbalance.
- The precision, which assesses the quality of positive predictions, measures how often a positive prediction is labeled positive.
- The recall, which measures the number of retrieved positives.
- And finally, the F1-Score, which combines the recall and precision scores.

Aside from assessing the quality of the binary predictions, measuring the quality of the direct output of autoencoders is also important. As illustrated in Figure 2, the state of the anomaly detection decision goes from continuous to binary using a thresholding technique, the Otsu thresholding. The thresholding performance then impacts the evaluation of the proposed pipeline after the thresholding step. A metric working on the continuous prediction product is thus an adequate complementary asset for evaluating the performance.

To alleviate this, we opt for the Receiver Operating Characteristic (ROC) curve and its induced Area Under the Curve (AUC) score.

### 3.3. Hyperparameter Settings

Deep learning algorithms have a dense set of hyperparameters to tune. To improve the repeatability of the presented results, we state the choices made regarding training parameters and embedding dimensions.

#### 3.3.1. Training Parameters and Setup

Various design choices through parameter settings need to be made to frame the training of neural networks. As we handle the training of 4 distinct models (FTCAE, CSTAE, and the two networks of the DSTAE), we detail the training parameters for each separately in Table 1.

**Table 1.** Training parameters per approach.

| Model | Learning Rate | Batch Size | Optimizer | Epoch Counts |
|---|---|---|---|---|
| FTAE | $1 \times 10^{-3}$ | 1024 | Adam | 20 |
| CSTAE | $1 \times 10^{-4}$ | 4096 | Adam | 20 |
| DSTAE (Spatial AE) | $1 \times 10^{-3}$ | 64 | Adam | 20 |
| DSTAE (1D-CAE) | $1 \times 10^{-3}$ | 1024 | Adam | 20 |

In addition, each network is equipped with a learning rate scheduler, which divides the learning rate by 10 on loss plateauing over five epochs, with a plateau delta set at a threshold of $1 \times 10^{-4}$. Thus, it is quite common during experiments to see at least one learning rate decrease for improved convergence.

#### 3.3.2. Impact of the Embedding Dimension

As mentioned earlier and illustrated in Figure 3, a critical design choice for the autoencoders is the bottlenecking/embedding dimension $t$. A dimension too big will make the reconstruction task too simplistic and will be detrimental to the convergence of the network. On the other hand, a dimension too small will be highly restrictive and may also impact the convergence of the autoencoder. No ideal dimension exists in the literature, and its discovery is often empirically optimized. For that matter, we trained the three separate methodologies using a variety of embedding sizes to enhance the understanding of its impact on fire retrieval.

For the FTAE network, as it ingests an input of shape $(29, 2) = 58$ values, we select embedding dimensions of size 1, 9, and 20. For the CSTAE approach, the input shape is $(7, 7, 29, 2) = 2842$ values. However, we consider the spatial values to be highly correlated. Thus, as the input information is expected to be redundant, we opt for more restrictive ratio-wise embedding dimensions of sizes 40, 100, and 200. For the spatial autoencoder of the DSTAE, given its inputs of dimensions $(128, 128, 2) = 32,768$, we select a fixed embedding dimension of 50. For the 1D convolutional autoencoder of the DSTAE, the spatial feature time series are of dimensions $(29, 8) = 232$ values. We then selected the following embedding dimensions: 4, 40, and 80.

## 4. Results

The presentation of the performances of the proposed pipeline, through the evaluation of various methodologies, is split into two parts. We first introduce the quantitative results using the five aforementioned metrics: accuracy, recall, precision, F1-Score, and AUC score. Then, we display the prediction score maps to visually analyze the methods' performance.

*4.1. Quantitative Analysis*

We introduce in Table 2 a numerical performance comparison between each method, alongside four supervised approaches, each trained using a stratified four-fold cross-validation scheme, repeated five times for statistical relevance of the presented results. We thus include the performance of the following models:

- Random Forest Classifier, parameterized with 100 trees.
- Logistic Regression, using an L2 penalty term.
- Quadratically Smoothed Support Vector Machine, with $\gamma$ set to 2.
- One-Dimensional Convolutional Neural Network, re-using the temporal encoder architecture introduced in Figure 3a.

**Table 2.** Performance comparison between the unsupervised FTAE, the CSTAE, and the DSTAE, as well as conventional supervised approaches for reference in the case where labelled training data are available. While usually performing better, supervised methods might not generalize well on different locations.

| | | Accuracy | Precision | Recall | F1-Score | AUC Score |
|---|---|---|---|---|---|---|
| FTAE | 1D Emb. | $0.88 \pm 0.01$ | $0.94 \pm 0.01$ | $0.52 \pm 0.01$ | $0.67 \pm 0.01$ | $0.90 \pm 0.01$ |
| | 9D Emb. | $0.87 \pm 0.02$ | $0.75 \pm 0.08$ | $0.65 \pm 0.03$ | $0.71 \pm 0.03$ | $0.90 \pm 0.01$ |
| | 20D Emb. | $0.89 \pm 0.01$ | $0.81 \pm 0.03$ | $0.72 \pm 0.01$ | $0.76 \pm 0.01$ | $0.93 \pm 0.07$ |
| CSTAE | 40D Emb. | $0.88 \pm 0.01$ | $0.86 \pm 0.01$ | $0.61 \pm 0.01$ | $0.71 \pm 0.01$ | $0.92 \pm 0.01$ |
| | 100D Emb. | $0.89 \pm 0.01$ | $0.87 \pm 0.01$ | $0.63 \pm 0.01$ | $0.73 \pm 0.01$ | $0.92 \pm 0.01$ |
| | 200D Emb. | $0.89 \pm 0.01$ | $0.86 \pm 0.01$ | $0.64 \pm 0.01$ | $0.74 \pm 0.01$ | $0.92 \pm 0.01$ |
| DSTAE | 4D Emb. | $0.80 \pm 0.01$ | $0.56 \pm 0.03$ | $0.73 \pm 0.01$ | $0.63 \pm 0.01$ | $0.85 \pm 0.01$ |
| | 40D Emb. | $0.89 \pm 0.07$ | $0.72 \pm 0.16$ | $0.72 \pm 0.05$ | $0.71 \pm 0.07$ | $0.89 \pm 0.04$ |
| | 80D Emb. | $0.86 \pm 0.03$ | $0.70 \pm 0.09$ | $0.79 \pm 0.04$ | $0.72 \pm 0.04$ | $0.90 \pm 0.01$ |
| Random Forest | | $0.90 \pm 0.01$ | $0.94 \pm 0.01$ | $0.64 \pm 0.01$ | $0.76 \pm 0.01$ | $0.93 \pm 0.01$ |
| Logistic Regression | | $0.91 \pm 0.01$ | $0.88 \pm 0.01$ | $0.73 \pm 0.01$ | $0.80 \pm 0.01$ | $0.93 \pm 0.01$ |
| SVM | | $0.91 \pm 0.01$ | $0.88 \pm 0.01$ | $0.72 \pm 0.01$ | $0.79 \pm 0.01$ | $0.93 \pm 0.01$ |
| 1D-CNN | | $0.90 \pm 0.01$ | $0.84 \pm 0.01$ | $0.69 \pm 0.01$ | $0.76 \pm 0.01$ | $0.91 \pm 0.01$ |

Best average performance within unsupervised and supervised methods are highlighted in green, while worst performance for unsupervised methods is highlighted in red.

Among unsupervised approaches, it appears that the fully temporal is the best performing overall, with the FTAE-20D configuration having the highest F1-score and AUC Score. However, the FTAE's performance is highly conditioned by the choice of the embedding dimension. While more precise, lower embedding dimensions offer lower recall regarding the addition of spatial information through the CSTAE and DSTAE approaches. At the same time, it does not solidly improve performance; the CSTAE metrics appear more stable, both within similar and across different embedding space configurations. This increase in robustness is added value for its operationality. Finally, while outputting acceptable performance, the DSTAE still falls behind the two other approaches regarding overall performance and robustness. The DSTAE appears to suffer from predicting too many false positives: while this strongly benefits his recall performance, its precision lacks behind as the worst among every method, no matter the tested configuration. We believe that it may be connected to a complexification of the information flow with the use of two decoupled autoencoders, which are not jointly optimized.

As expectable, supervised approaches reach the best performances overall. However, our unsupervised approach is on par with these reference methods as shown by FTAE results of 93% AUC Score (best score) or 0.76 F1-Score on par with random forests or 1D-CNNs. This is obtained (1) without the need for training data, and has by design (2) better generalization properties as supervised methods depend on the data seen at training time.

## 4.2. Qualitative Analysis

We display in Figure 6 the prediction maps of the three unsupervised and supervised approaches: it appears clear that they distinguish the forest fire outline better from the background, with a much larger contrast. However, this is expected as the supervised approaches maximize the prediction probability value of fire areas and minimize the prediction probability value of non-fire areas.

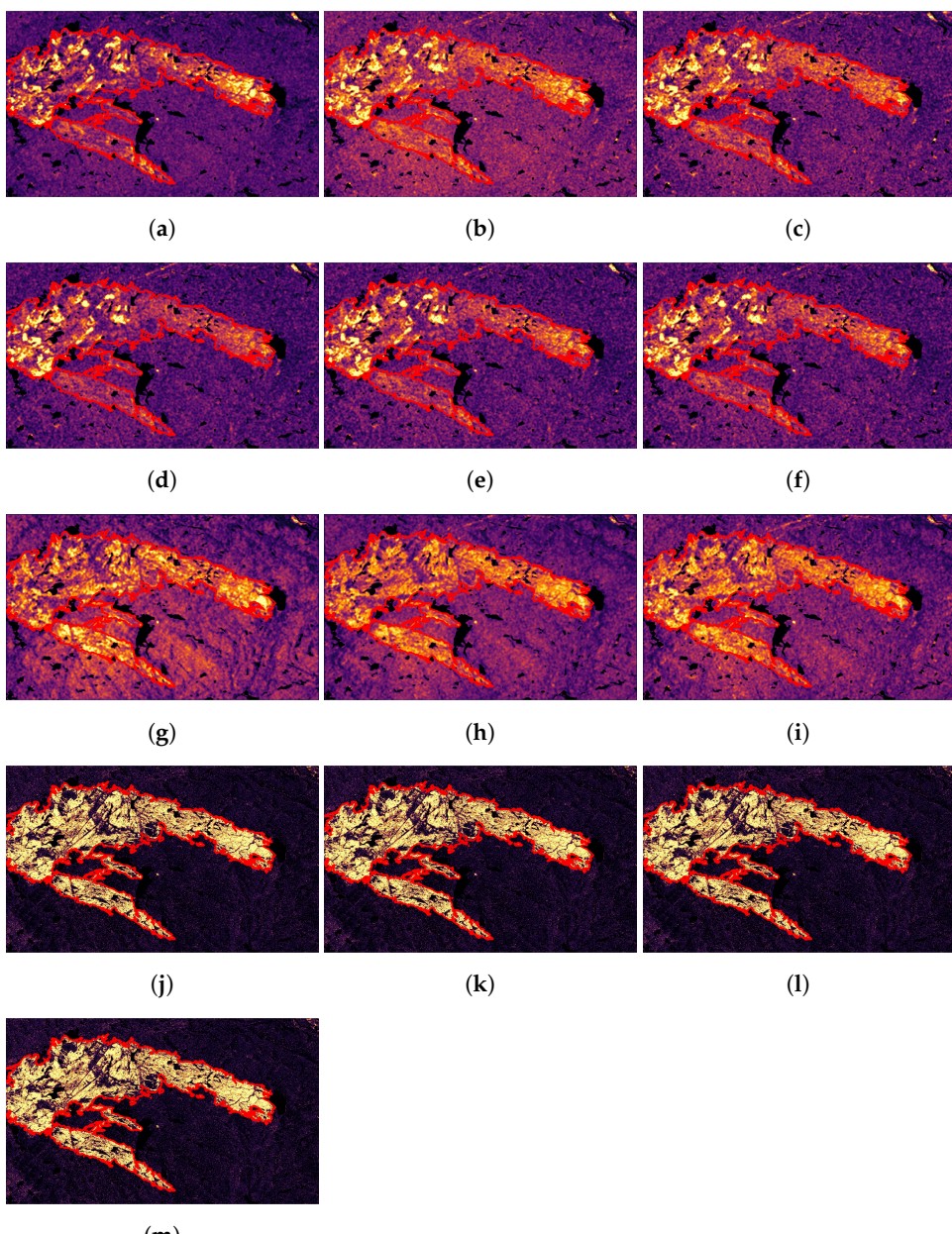

**Figure 6.** Visual comparison of the continuous average $\Delta MSE_{smoothed}$ maps of the three unsupervised methods, as well as the prediction probability maps of the four supervised approaches, with the ground truth (red contour) (**a**) FTAE-1D, (**b**) FTAE-9D, (**c**) FTAE-20D, (**d**) CSTAE-40D, (**e**) CSTAE-100D, (**f**) CSTAE-200D, (**g**) DSTAE-4D, (**h**) DSTAE-40D, (**i**) DSTAE-80D, (**j**) Random Forest, (**k**) Logistic Regression, (**l**) SVM, and (**m**) 1D-CNN.

When focusing the visual analysis on the comparison of the unsupervised approaches, while the approximate shape of the fire outline is visible in each of the nine images, its exact contour changes depending on the used approach and the embedding dimension selected. For instance, the FTAE approach appears to highlight more and more pixels

as fire pixels the bigger the embedding dimension, which explains the precision/recall trade-off observed in the performance metrics. In addition, it appears that the fire areas are heterogeneously colored. While not investigated, we theorize that these variations may correlate with the remaining trees' state. This heterogeneity also appears in the CSTAE maps, which are much more similar across embedding configurations, again in a similar fashion as previously illustrated in the numerical performances. Finally, while the DSTAE has the lowest F1 and AUC scores, its prediction maps highlight the expected fire outline. In particular, in Figure 6h,i, the outline is notably well-detailed, visible, and much more homogeneous than other approaches. These results favor the combined use of the three introduced methods.

### 4.3. Runtime Analysis

Due to their intricate design differences, the runtime of each method is different and plays a significant role in the choice of the retained approach. For that, the duration of the entire pipeline for each tested approach is introduced in Table 3. The pipeline is split between the training and inference time. For the unsupervised approaches, the training time incorporates fitting the autoencoders. In contrast, the inference time comprises the generation of the MSE images, their difference, the smoothing, and finally of the Otsu thresholding step. The total time thus corresponds to the time required to obtain a disturbance prediction from scratch data. The experiments were run on a machine equipped with an i7-10700k, 64 GB of RAM, and a RTX 3090 GPU.

**Table 3.** Runtime comparison between the unsupervised FTAE, the CSTAE, and the DSTAE, as well as conventional supervised approaches.

| | | Training (in min) | Inference (in min) | Total (in min) |
|---|---|---|---|---|
| FTAE | 1D Emb. | 26 | 4 | **30** |
| | 9D Emb. | 26 | 4 | **30** |
| | 20D Emb. | 26 | 4 | **30** |
| CSTAE | 40D Emb. | 332 | 15 | **347** |
| | 100D Emb. | 349 | 17 | **366** |
| | 200D Emb. | 360 | 17 | **377** |
| DSTAE | 4D Emb. | 36 | 4 | **40** |
| | 40D Emb. | 36 | 4 | **40** |
| | 80D Emb. | 36 | 4 | **40** |
| Random Forest | | 22 | 1 | **23** |
| Logistic Regression | | 27 | <1 | **27** |
| SVM | | 3 | 1 | **4** |
| 1D-CNN | | 175 | 2 | **177** |

As expected, the most straightforward approach, the FTAE, performs the fastest among unsupervised methodologies, with a total algorithm run time of 30 min, against 40 for the DSTAE and more than 300 for the CSTAE. Unsurprisingly, the added spatial information of 3D inputs drastically increases computation time for little to no performance gain. In addition, compared to supervised approaches, the FTAE runs slightly slower than its non-deep learning counterparts, but with the added benefit of being fully unsupervised. For that matter, with performances and computation time on par with supervised learning approaches, while not requiring ground truth data for its training, the FTAE approach is ideal for the proposed monitoring pipeline.

## 5. Discussion

Considering the promising performance of the developed methodology, in particular, in comparison to supervised approaches, we believe that the pipeline presented in Figure 2 has the operational potential for monitoring AOIs against fires. An imagined use-case

stands as follows: a given authority requires continuous annual monitoring of a specific forested area under any weather condition. For that matter, a monitoring system plugged into the latest Sentinel-1 imagery would compare the current period's temporal signature to the previous period's temporal signature, considered a baseline. Any changes would then be annotated, and a new model would be trained on the time series of the current period without the locations deemed anomalous. This iterative process would thus allow for continuous monitoring without the need for human intervention. The regular retraining would likewise help to alleviate a common problem of deployed machine learning models: Concept Drift [40].

In addition, while presented in the context of forest monitoring, this methodology can detect period-on-period changes in any vegetation body with a seasonal and regular signature. Its extension to the fields of agricultural and land monitoring is thus envisaged. However, this expectation may also be the limitation of the methodology: any non-seasonally regular environment will not benefit from leveraging the reference period temporal signature to detect changes in the monitoring period. Such changes may better be isolated using purely spatial approaches, in contrast with the purely temporal and spatiotemporal approaches introduced in this work. Another limitation comes from the change detection capacities of the Sentinel-1 satellites: equipped with C-Band SAR sensors, subtler changes, such as losses in pine thorns, may not impact the backscatter: for that matter, the addition of SAR imagery measured at a different wavelength, or of different sensor technologies (optical, infrared...) may prove itself crucial. Thus, a multi-sensor autoencoding approach can be imagined, leveraging the all-weather capacities of Sentinel-1 while retaining a complete description of the scene through the use of other sensors.

## 6. Conclusions

This work presents a methodology for monitoring a forest AOI against fires using Convolutional Autoencoders and Sentinel-1 time series. The autoencoders are used to model a given AOI's expected temporal SAR signature by training on a reference period before being applied to the SAR time series of a monitoring period. With the addition of spatial smoothing and Otsu thresholding, the calculation of the reconstruction error of the autoencoder allows for the detection of period-on-period changes. We propose three different autoencoding architectures, from fully temporal to spatiotemporal approaches, and we compare their performance with various latent space dimensions. We also compare the introduced methods to commonly used supervised learning algorithms to estimate their relevance. While these supervised methods outperform the proposed architecture, the performance gap is slight. When considering the added benefits of unsupervised learning training, namely the absence of any label and label-induced noise, and increased robustness to time or issues of domain shift, an unsupervised learning approach appears preferable for applicative scenarios, as introduced in this paper. In particular, their application to period-on-period monitoring leaves room for a continuous learning framework for early detection of fires. It can then be autonomously trained, deployed, and maintained, thus fitting the needs for fully automated pipelines, needing as little as possible of human intervention.

**Author Contributions:** Conceptualization, Thomas Di Martino and Bertrand Le Saux; methodology, Thomas Di Martino and Bertrand Le Saux; software, Thomas Di Martino; validation, Thomas Di Martino; formal analysis, Thomas Di Martino and Bertrand Le Saux; investigation, Thomas Di Martino; resources, Thomas Di Martino and Bertrand Le Saux; data curation, Thomas Di Martino; writing—original draft preparation, Thomas Di Martino; writing—review and editing, Thomas Di Martino, Bertrand Le Saux, Régis Guinvarc'h, Laetitia Thirion-Lefevre and Elise Colin; visualization, Thomas Di Martino; supervision, Bertrand Le Saux, Régis Guinvarc'h, Laetitia Thirion-Lefevre and Elise Colin. All authors have read and agreed to the published version of the manuscript.

**Funding:** This research received no external funding.

**Data Availability Statement:** Data for the experiments are publicly available. Authors may be solicited to transmit them, or guide one towards their obtention.

**Acknowledgments:** The authors would like to thank Natural Resources Canada for their Open Data policy, which made the experimentation possible. In addition, Thomas Di Martino would like to thank Bertrand Le Saux, Pierre-Philippe Mathieu, and every colleague from the ESA Φ-lab for their scientific support in the development of this study, as well as their warm welcome.

**Conflicts of Interest:** The authors declare no conflict of interest.

## Abbreviations

The following abbreviations are used in this manuscript:

| | |
|---|---|
| AOI | Area Of Interest |
| AE | AutoEncoder |
| CSTAE | Coupled SpatioTemporal Autoencoder |
| DCSTAE | DeCoupled SpatioTemporal Autoencoder |
| ELU | Exponential Linear Unit |
| ERS | European Remote Sensing |
| FTAE | Fully Temporal Autoencoder |
| MSE | Mean Squared Error |
| ROC | Receiver Operating Characteristic |
| SAR | Synthetic Aperture Radar |

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
