# Peer review of "Detection of Forest Fires through Deep Unsupervised Learning Modeling of Sentinel-1 Time Series"

_ijgi, doi:10.3390/ijgi12080332_

Round 1

Reviewer 1 Report

Full Title: Monitoring of Forested Area of Interest through Deep Unsupervised Learning modeling of Sentinel-1 time series ijgi-2493831

General considerations

The authors’ proposed methodology and its application to SAR image time series is innovative. In the reviewer’s opinion, the paper is finally worthy of publication. The research could potentially be strengthened by conducting a sensitivity analysis of various ecosystem parameters and validating the results with in situ data, both in terms of fire dates and the surfaces and perimeters of fire-affected areas.

Title

While the proposed method has the potential to be applied to various objectives, it has only been tested and verified for detecting forest areas affected by fire. The current title does not accurately reflect the research topic. Therefore, the reviewer suggests that the authors either test the method for other objectives or revise the text to emphasize the specific application that has been tested.

Specific comment

Line 66: “General consideration”: The methodology section appears to lack a verification with field data, at least the perimeters and surfaces of the areas covered by fire; data that are typically acquired by firefighting agencies. Furthermore, the manuscript could be strengthen by a sensitivity analysis at least for the most important fire mapping parameters including seasonality and frequency. Even though soil pedology, vegetation types and post-fire recovery vegetation play a primary role in distinguishing the burnt surface from remote sensing and in its delimitation.

Line 112: “Otsu thresholding”: Well, the threshold is determined by minimizing intra-class intensity variance, or equivalently, by maximizing inter-class variance. The two classes are to be intended as two periods for a given pixel before and after the disturbance?

If so, how the algoritm behaves if two dirsturbances occur at two different time over the time series?

For example, two fires over the same pixel in two different times.

Line 140: “7-by-7 neighborhood: I was wondering if the size of the kernel should be a function of the extension of the fire.

Line 181: “Ascending mode”: The method is applied on images acquired during the ascending orbit. It can also be independently applied to images recorded during the descending orbit. This could provide confirmation of the timing of the fires. Also, it would be interesting to see if there are any differences in deviations due to the different viewing geometry of ascending and descending orbits.

Table 2: please modify as reported in the file

Line 271: “overall Performance”: It’s worth noting that the highest recall score was achieved by the DSTAE 80D Emb., surpassing the scores of conventional approaches that typically perform well in most metrics.

At the same time the DSTAE 4D Emb shows often the worst performances.

Line 275: “best performances overall”: It appears that Logistic Regression achieves the best overall performance. I’m curious as to why this is the case.

Figure 5: “prediction maps”: I wonder about the smallest size of a burnt area that the method is able of accurately identifying by using Sentinel-1 time series.

Line 284: “more pixels as fire pixels the bigger the embedding dimension”: Do the authors have any field data that can be used to verify the accuracy of the fire perimeters?

I will be available to authors for the assessment of the manuscript at the subsequent submission.

Best regards

Reviewer 2 Report

Dear Authors

Please redesign and rewrite the mnuscript.

Bests,

Reviewer 3 Report

Overall Impression
In this manuscript, the authors propose several methods of deep learning to detect boreal forest disturbances, namely fires, using Sentinel-1 SAR data. In principle, the approach is similar to any number of trajectory-based change detection algorithms: train the algorithm on a reference period and get an expected time series, then compare that expected time series against model outputs. The authors’ use of an autoencoder (AE) algorithm delegates the generation of the expected series to machine learning, and the comparison relies on the degree of difference between mean squared errors for reference and monitored years. (This means that the AE need not model the actual time series objectively well; it simply has to detect when the model quality shifts.)

While the manuscript is interesting, very timely, and does demonstrate the promise of AEs, there are several areas that can and should be improved. Firstly, the authors need to better highlight the benefits of unsupervised approaches: they mention the lack of labels (which can constrain supervised approaches) and describe a conceptual situation in the discussion, but a more detailed example in the introduction would be compelling. Secondly, the use of AEs and deep learning is outside my field of expertise, and I suspect it will be novel to much of the readership. The more the authors make the AEs and their use accessible, the better for the manuscript. (Examples are in the general comments.)

If the authors address these and more minor quibbles (see line-by-line comments), I would be happy to reconsider the manuscript.

General Comments
While the authors explain the idea of AEs reasonably well (Figure 3 is a great help), I am not qualified to judge the specifics and parameters of the AEs used.

A brief description of Ostu thresholding would be helpful. Likewise, I understood the basic idea of bottlenecking, but anything like an application/example would be very helpful, because it’s not clear how a time series is being decomposed to a lower-dimensional code (I keep imagining a compressed digital file), then being restored.

In Table 2 and other figures/tables, it would be very helpful to assign a more intuitive tag to each AE, as I expect the abbreviations will mean little to the average reader. For example, “spatial and temporal” to CSTAE, and “spatial, then temporal” to DSTAE.

Figure 5 shows prediction maps, but where is the reference by which the reader can compare? We’re given a fire boundary in Figure 4, but it isn’t clear that the entirety of the area within was burned, or burned with uniform severity. At the very least, please add the outline to the Figure 5 maps.

What is the time/computing cost for these AEs, as opposed to supervised algorithms? Clearly they require less human interaction, but are they comparably efficient in terms of processing?

Line by Line Comments
18: It seems odd to say the first environmental monitoring satellite was launched 1991, given the widespread use of Landsat TM data dating back to 1984 (or earlier Landsat) for this purpose. I recognize the difference in designs, but the applications are well-documented.

23: Suite, instead of suit?

25: Again, “bleeding edge” seems an odd descriptor for a constellation of satellites for which the most recent launch was almost four years ago. Also, the term seems colloquial.

305: “regular” instead of “regularly”

Round 2

Reviewer 1 Report

Dear Authors,

all my requests for enlightenment have been appropriately addressed in the answers and in the text.

In my opinion I accept in this form.

Best regards